# Safety Impact Assessment of Optimal RWIS Networks—An Empirical Examination

**Simita Biswas** [1,*], **Davesh Sharma** [2] **and Tae J. Kwon** [1]

1    Department of Civil and Environmental Engineering, University of Alberta, Edmonton, AB T6G 2W2, Canada
2    EXP, Burnaby, BC V5G 4W3, Canada
*    Correspondence: simita@ualberta.ca

**Abstract:** Optimal RWIS network can be defined as an RWIS configuration where the total number of stations (RWIS density) are determined based on a well-established guideline and the locations are allocated systematically assuming that it will provide the maximum monitoring coverage of the network. This paper examines and quantifies the benefit of an optimized RWIS network and how these benefits impact traffic safety. The methodological framework presented herein builds upon our previous efforts in RWIS location-allocation, where the kriging variance is used as a performance indicator for monitoring coverage. In this study, the network coverage index (*NCI*) parameter is proposed to gauge RWIS network performance and quantitatively evaluate its impact on traffic safety. The findings of this study reveal a strong dependency between the *NCI* and the RWIS network configuration. In terms of traffic safety, the relationship between *NCI* and safety effectiveness can be expressed as a polynomial function, where the two are proportional to one another. In the state of Iowa, an RWIS network with 80% monitoring coverage (*NCI* = 0.8) can reduce additional 40 collisions per site annually compared to a network without RWIS stations. Based on the findings obtained in this study, road agencies and RWIS planners can now be assisted with conceptualizing the capabilities of an optimized RWIS network, which will help them increase monitoring coverage, and in the process, gain a quantitative understanding on its potential impact on traffic safety.

**Keywords:** Road Weather Information System (RWIS); impact assessment; optimal RWIS network; safety evaluation; ordinary kriging; error variance

## 1. Introduction and Background

According to a recent study by the Federal Highway Administration (FHWA), adverse weather condition causes about 21% of all road collisions every year in the U.S. [1]. Statistics shows that, over 1.5 million road accidents, 0.8 million injuries, and 7000 fatalities occur annually in the U.S. due to adverse weather [2]. In its northern neighbor—Canada, every year, about 3000 deaths result from weather-related road crashes, and over 1 in 135 people experience driving-related injuries [3]. According to the Ontario Road Safety Annual Reports, poor road surface condition leads to 17 percent increase in vehicle crashes over the last 16 years [4]. As a result, winter-related road incidents have become a significant concern for many jurisdictions. Among many strategies that exist to meet this objective, one approach is to provide better road conditions through more efficient maintenance operations. This method would require maintenance personnel to thoroughly understand both weather and road conditions on their road network, which can be done through one of the most critical pieces of highway intelligent transportation systems (ITS) infrastructure called Road Weather Information Systems (RWIS).

RWIS consist of a group of sensors that monitors road weather and surface conditions along the road network. Information regarding the road weather and surface condition are collected, processed and disseminated by the collaboration of advanced sensors. The collected information is used by road maintenance authorities to make timely operative decisions aimed at improving traffic safety and mobility before, during, and after inclement

weather events. Additionally, RWIS provides travelers with better information via RWIS-connected dynamic message signs to help them make more informed travel decisions, which can reduce weather-related crashes and injuries [5,6]. Despite their usefulness and numerous benefits, a few limitations associated with RWIS stations have been identified thereby hindering their distribution. The most significant limitation is the installation cost. Depending on the type and number of sensors equipped, the cost could be as high as $100,000 U.S. dollars per station [7,8]. Considering the limited budget for RWIS deployment and the random nature of road weather fluctuations, RWIS stations must be implemented strategically to ensure optimal monitoring coverage under varying circumstances to ensure both mobility and safety.

Due to the importance of RWIS infrastructure, numerous studies have been conducted to establish a siting guideline for RWIS installation. An extensive research was conducted by the U.S. Federal Highway Administration (FHWA) in 2005, where they analyzed existing information and conducted interviews with several state Department of Transportations (DOTs). According to this study, based on the knowledge and experience of field operators, 30 to 50 km (20 to 30 miles) of spacing is recommended between RWIS stations [8]. As the recommended guidelines were based on experts' opinion and prior experiences, several researches were conducted to identify a more systematic way to quantify the spatial coverage of RWIS data and optimal placement of RWIS stations [2,9–13].

In a GIS-based study conducted by Kwon and Fu (2013), a framework for RWIS network location evaluation was presented, where the variability of the surface temperature (VST), mean surface temperature (MST), and snow water equivalent (SWE) were considered alongside topographical location attributes. The output of this study revealed the feasibility of developing a systematic process for RWIS installation using an integrated location criterion by capturing multiple variables [14]. Similarly, in a more recent study conducted by Kwon et al. (2017), RWIS network location optimization was performed through an innovative geostatistical analysis technique—kriging. Optimization was formulated as a Nonlinear Integer Programming (NIP) problem to maximize the monitoring capability while minimizing the spatially averaged kriging variance of hazardous road surface conditions. The RWIS data used in this study were taken from the state of Minnesota, U.S., to evaluate the effectiveness of the current RWIS location setting and make recommendations for future network expansion. Although the method developed contributed to delineating RWIS locations, it only dealt with a spatial domain without considering inherent temporal variations of road weather parameters, making their location solutions less conclusive [15]. Hence, Kwon and Fu (2017) further extended their previous work to investigate the dependency of optimal RWIS spacing of a region on the spatiotemporal variability of road weather conditions and corresponding topographical characteristics of the region. A group of case studies were conducted by the authors using data from three U.S. states (Iowa, Utah, and Minnesota) and one Canadian province (Ontario) [16]. Although the output of this research entails that the number of RWIS stations required for a region would depend on the topographic settings, they did not provide a systematic method for regions with limited or no RWIS information.

Considering the resurgent need to build a systematic and generalized RWIS network planning guideline, our previous efforts have developed a comprehensive and transferable methodological framework to optimize the design of a regional RWIS network by incorporating the use of multiple RWIS variables for improved spatiotemporal inference [17–19]. Two crucial network planning questions were answered through that research: (i) how many RWIS stations are needed in a region with varying environments to provide sufficient coverage over space and time? And (ii) where should these stations be located to provide adequate monitoring coverage of a given region? In other words, the optimal density and location of RWIS stations were determined to provide the maximum monitoring coverage for a region. During the problem formulation, it was assumed that the optimal RWIS network provides the maximum coverage of the region under investigation. Geostatistical approaches were implemented to determine the optimum number of RWIS stations across

several topographic and weather zones covering 14 U.S. States. In the subsequent step, a methodological framework was developed to determine the optimal RWIS locations by considering the spatiotemporal characteristics of critical RWIS variables.

Although the generalized guideline developed in our previous studies provides a solid foundation for RWIS network planning, no prior efforts have been made to examine and quantify the benefits of an optimally situated RWIS network. While developing the optimal RWIS location solution, it was implicitly assumed that each solution set is associated with a unique spatial configuration tied to an objective function value or sum of kriging variance that represents RWIS' monitoring capability. The solution set associated with the lowest objective function value (lowest kriging variance) would be considered the solution with the highest network coverage and thus assumed to be most beneficial [19]. Based on this presumption that network coverage is a vital parameter for determining the goodness of an RWIS configuration, there is a resurgent need to extend this effort by investigating if it could also be used to explain its impact on traffic safety—a worthwhile attempt that has never been made in existing literature pertaining to quantifying the safety benefits of RWIS location solutions.

Therefore, the primary objectives of this study are: (a) to investigate the relationship between a newly created measure called network coverage index (*NCI*) and network configuration of RWIS, and (b) quantitatively assess the impact of *NCI* on the transportation system based on collision reduction potential. The findings of this research will provide a clearer understanding of the benefit of an optimal RWIS solution and its impact on the transportation system.

The remainder of the paper is organized as follows. The next section presents the methodological approach used to meet the research objective, followed by a description of the study area and data used. Next, the results obtained are analyzed in detail with discussion. This is succeeded by the Section 5, which highlights the key findings and provides recommendations for future research.

## 2. Methodology

### 2.1. Overview of Research Procedures

The first phase of this study was the database development by aggregating and integrating various data sets into GIS. Two datasets were developed, one to determine the *NCI* (a more detailed description is to follow) and another to evaluate safety.

After extracting the RWIS station data, a quality check was performed using the following steps: data completeness test, reasonable range test, and a neighborhood value comparison. Following this, detrending was performed with respect to time using Generalized additive model (GAM) [20,21], followed by geostatistical analysis. Spatiotemporal analysis was performed by constructing empirical semivariograms from the processed data, which optimizes parameter estimations for unsampled locations and captures the possible autocorrelation associated with the RWIS variables. Joint semivariogram models were then developed by combining spatial and temporal semivariograms to evaluate the spatiotemporal variability of RWIS measurements [19]. Based on parameters obtained from the joint semivariogram, kriging interpolation was used to estimate values at unsampled locations and their estimation error or kriging variance. Kriging variance was then utilized to determine the *NCI* for respective RWIS networks. The procedure was repeated for each set of RWIS configurations to investigate its impact on the *NCI*.

In terms of safety evaluation, 12 years (2008 to 2019) of inclement winter weather collision data were extracted, among which collisions due to poor road surface conditions were isolated for safety evaluation. Additionally, only major network roads, i.e., Interstate, State, and U.S. highways were considered due to maintenance departments prioritizing major roads for treatment. RWIS stations included in the safety evaluation were selected based on three review criteria: (a) data review to ensure that sufficient before and after period collision data were available, (b) geometry review to ensure that no major design nor construction activities occurred near the RWIS stations, and (c) operation review to ensure

that minimal operation gaps were present in the data. The processed data were employed to calibrate the safety performance functions (SPFs) and yearly calibration functions (YCFs). Next, Empirical Bayes (E.B.) analysis was applied to determine the collision reduction associated with each of the selected RWIS stations [22].

Upon processing the data, the impact of *NCI* on collision reduction was assessed in order to evaluate the goodness of the RWIS location solutions. Research procedures for this study are summarized in Figure 1.

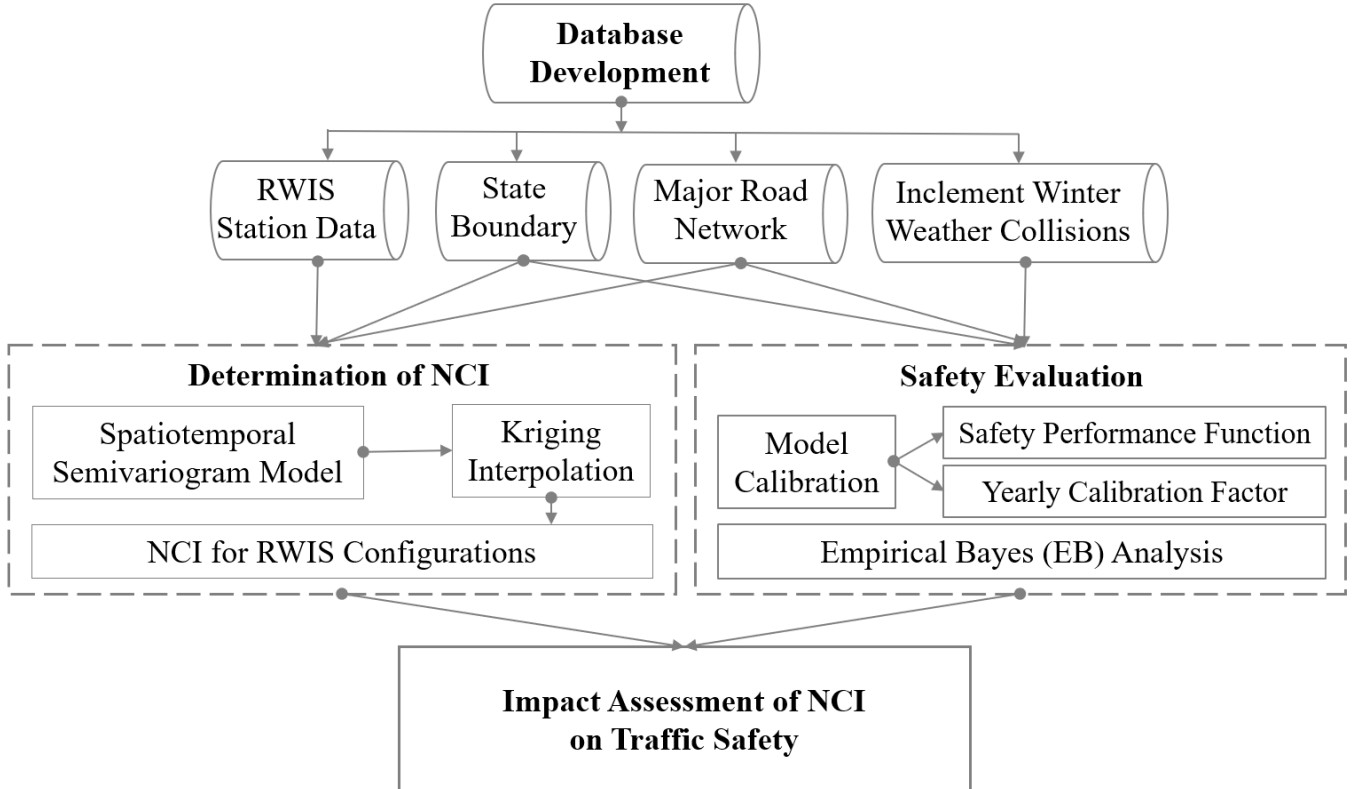

**Figure 1.** Methodological flowchart.

### 2.2. Determination of Network Coverage Index (NCI)

The network coverage index (*NCI*) was used to rate the monitoring capabilities of a defined RWIS configuration for a specific region. It is a surrogate measure that ranges between 0 and 1, where 0 represents no monitoring coverage, and 1 represents complete network coverage.

Determining *NCI* requires the use of kriging, which is a widely used geostatistical technique that provides the best linear unbiased estimate (BLUE) for variables that vary over space [23]. The weighted average of the observed data was used in kriging to predict values at unsampled locations, where the weights were determined based on the separation distance between the sampled points and unsampled locations. Kriging provides estimates at unknown locations along with estimation errors by quantifying the spatial variability over the area of interest [24]. Ordinary Kriging (OK) is a form of kriging that assumes the mean to be unknown but constant over each local neighborhood [24,25]. OK estimation variance for an estimation location, $x_0$ can be defined by the following equation:

$$\sigma_{OK}^2(x_0) = g'G^{-1}g \tag{1}$$

where, $G$ is the semivariance matrix between the observations and $g$ is the semivariance matrix between observations and unsampled points. The equations of $G$ and $g$ are given below:

$$G = \begin{bmatrix} \gamma(x_1, x_1) & \gamma(x_2, x_1) & \dots & \gamma(x_k, x_1) & 1 \\ \gamma(x_1, x_2) & \gamma(x_2, x_2) & \dots & \gamma(x_k, x_2) & 1 \\ & & \dots & & \\ \gamma(x_1, x_k) & \gamma(x_2, x_k) & \dots & \gamma(x_k, x_k) & 1 \\ 1 & 1 & \dots & 1 & 0 \end{bmatrix} \text{ and } g = [\gamma(x_0, x_1) \ \gamma(x_0, x_2) \dots \gamma(x_0, x_k) \ 1]' \tag{2}$$

Here, $x_i$ ($i = 1, 2, \dots, k$) is the sampling site of a sample subset of size $k$, where $k$ = number of RWIS stations. $\gamma(x_i, x_j)$ is the semivariance between sampling site $i$ and $j$.

Semivariance values were calculated by constructing empirical semivariograms from RWIS measurements. A semivariogram depicts the spatial autocorrelation between measured data points. It is a statistic that determines the similarity between two measurements as a function of separation distance [26]. Semivariance values are calculated by taking the average squared differences between two measured data points in a spatial domain separated by a defined lag distance. The general equation of semivariance estimation is presented in Equation (3).

$$\gamma(h) = \frac{1}{2n(h)} \sum_{i=1}^{n(h)} [z(x_i + h) - z(x_i)]^2 \tag{3}$$

Here, $\gamma(h)$ is the estimated semivariance; $z(x_i + h)$ and $z(x_i)$ are two measurements taken at location $x_i$ and $(x_i + h)$ separated by a lag distance $h$.

Since RWIS measurements (i.e., road weather variables) vary over both space and time, spatiotemporal semivariogram models are employed in this study. Both spatial and temporal autocorrelation associated with the RWIS variables are well captured in spatiotemporal modeling that optimizes parameter estimations for unknown locations. A set of variables, $z$ in a spatiotemporal field can be defined as a combination of spatial domain (S) and temporal domain (T): $z = \{z(s, t) | s \ \epsilon \ S, \ t \ \epsilon \ T\}$. The general equation of a random field $Z$ can be defined as: $z_i = Z(s, t)$, $i = 1, 2, 3, \dots n \times T$. Here, $n$ = number of sampled locations and $T$ = number of points in time. The most common formula for spatiotemporal semivariance estimation is shown in Equation (4).

$$\gamma(h_s, h_t) = \frac{1}{2n(h_s, h_t)} \sum_{k=1}^{n(h_s, h_t)} [z(s_k, t_k) - z(s_k + h_s, t_k + h_t)]^2 \tag{4}$$

Here, $\gamma(h_s, h_t)$ is the estimated semivariance, $n(h_s, h_t)$ is the total number of pairs in the random field, $z(s_k, t_k)$ is the observation at location $s_k$ and temporal point $t_k$, $z(s_k + h_s, t_k + h_t)$ is another observation at location $(s_k + h_s)$ and temporal point $(t_k + h_t)$. The observations pairs are separated by a user-defined spatial lag $(h_s)$ and temporal lag $(h_t)$ [18,27–29]. Spatial and temporal semivariograms can be combined using spatiotemporal anisotropy to estimate the joint semivariogram that can preserve both spatial and temporal effect. The joint semivariogram models developed in our previous effort are adopted in this study for conducting kriging interpolation [19].

Based on the formulation shown in Equation (1), we can drive *NCI* using estimation error (kriging variance) under the assumption that a higher estimation error represents an increased need for an RWIS station. Since the optimization is aimed at finding locations that minimize total estimation error, this would mean the optimal RWIS network provides the best monitoring coverage because it has the lowest estimation error. Therefore, the

kriging variance is inversely proportional to *NCI*, and the estimation error can be translated into *NCI* using the following equation.

$$NCI = \frac{K}{Kriging\ variance} \tag{5}$$

Here, *K* = Proportional factor. The value of *K* is sensitive to the regional attributes and can be established by determining the kriging variance at optimal conditions.

### 2.3. Safety Evaluation of RWIS Network

The collision reduction factor or the percent reduction in collisions was estimated in our previous efforts using the state-of-the-art before-and-after Empirical Bayes (E.B.) method [22]. E.B. accounts for the Regression-to-the-Mean (RTM) artifact by incorporating two separate pieces of information; (i) the collision history of the treatment sites and (ii) their predicted collision frequencies obtained from the Safety Performance Function (SPF)s. The ratio of the observed and expected number of collisions in the post-implementation period is the collision reduction factor of the countermeasure. Generally, two clues are used in EB method. The first one is the collisions that have already occurred at the treatment site and the second one is a set of reference sites that are similar to the treatment sites, so that it can represent the scenario as to what would have happened at the treatment site if the treatment not been implemented. Here, the treatment sites can be described as the sites that are within the influence region of RWIS station and the reference sites are those that are outside the influence of any RWIS station. Data from reference sites are used for local calibration of SPF and YCF. The overall process was divided into three steps. First, the expected collision frequency in the before period was estimated using the following equations.

$$N_{Expected,B} = w \times N_{Predicted,B} + (1 - w) \times N_{Observed,B} \tag{6}$$

$$w = \frac{1}{(1 + k \times N_{Predicted,B})} \tag{7}$$

where, $N_{Expected,B}$ is the expected collision frequency in the before-period, $w$ is the weighted adjustment factor between 0 to 1, $N_{Predicted,B}$ is the predicted collision frequency is the before-period, $N_{Observed,B}$ is the observed collision frequency in the before-period, and $k$ is the negative binomial overdispersion parameter estimated from SPF. A weighted sum of two separate pieces of information was used in this step. The predicted collisions for each site in the before-period were calculated using the calibrated SPF equation. In contrast, the observed collision frequencies come directly from the collected dataset. The equation of SPF adopted for safety evaluation is shown below:

$$\mu = e^{(\beta_0)} L^{(\beta_1)} \cdot V^{(\beta_2)} \tag{8}$$

Using this equation, the collision frequency in the before-period ($\mu$) can be predicted using road length (L) and traffic volume (V). Here, $\beta_0$, $\beta_1$ and $\beta_2$ are the regression parameters. The calibrated SPFs were developed in our previous study for several RWIS stations in Iowa [22]. Additionally, there are various confounding factors, such as improvements in the roadway, general traffic safety trends, and changes in weather conditions that cannot be captured by the SPFs. Therefore, Yearly Calibration Factors (YCFs) were also incorporated into the safety evaluation process. YCF can be defined as the ratio between the sum of observed collision frequencies and the sum of predicted collisions for the reference sites.

In the second step, the expected collision frequencies in the after period, $N_{Expected,A}$ was calculated. The calibrated SPF equations were used to estimate an adjustment factor, *Adj*, that captures the traffic volume variations during the before and after periods.

$$N_{Expected,A} = N_{Expected,B} \times Adj \tag{9}$$

The last step involves estimating the effectiveness of the countermeasure. The safety effectiveness or percent collision reduction due to countermeasure implementation was estimated using the following equation. Note that an odds ratio was used to account for potential bias.

$$Percent\ Collision\ Reduction = 100 \times (1 - Odds\ Ratio) \tag{10}$$

### 2.4. Impact Assessment of RWIS Network on Traffic Safety

After quantitatively measuring the impact of RWIS stations on traffic safety, the benefit associated with an RWIS configuration can be evaluated by assessing *NCI*. By utilizing the safety evaluation output, the collision reduction for each RWIS station can be calculated. The values of *NCI* for a set of RWIS network configurations were then plotted against the number of collision reduction for corresponding RWIS setup. Here, collision reduction was used as a performance indicator of safety benefits. Higher number of collision reduction is expected to be associated with an RWIS network of lower kriging variance, thus higher *NCI* value.

## 3. Study Area and Data

### 3.1. Study Area

Iowa—the selected study area—is a flatland region consisting of rolling plains and flat prairies, with altitudes ranging from 146 m to 509 m above sea level. This state was categorized as a moderate-severe weather region [17] where the adverse winter negatively impacts the transportation system. In regions like this, RWIS information plays a critical role, where the information it provides increases the responsiveness of winter road maintenance activity. The RWIS network of this state consists of 86 stations. Figure 2 represents the distribution of RWIS stations along with the major road networks in Iowa.

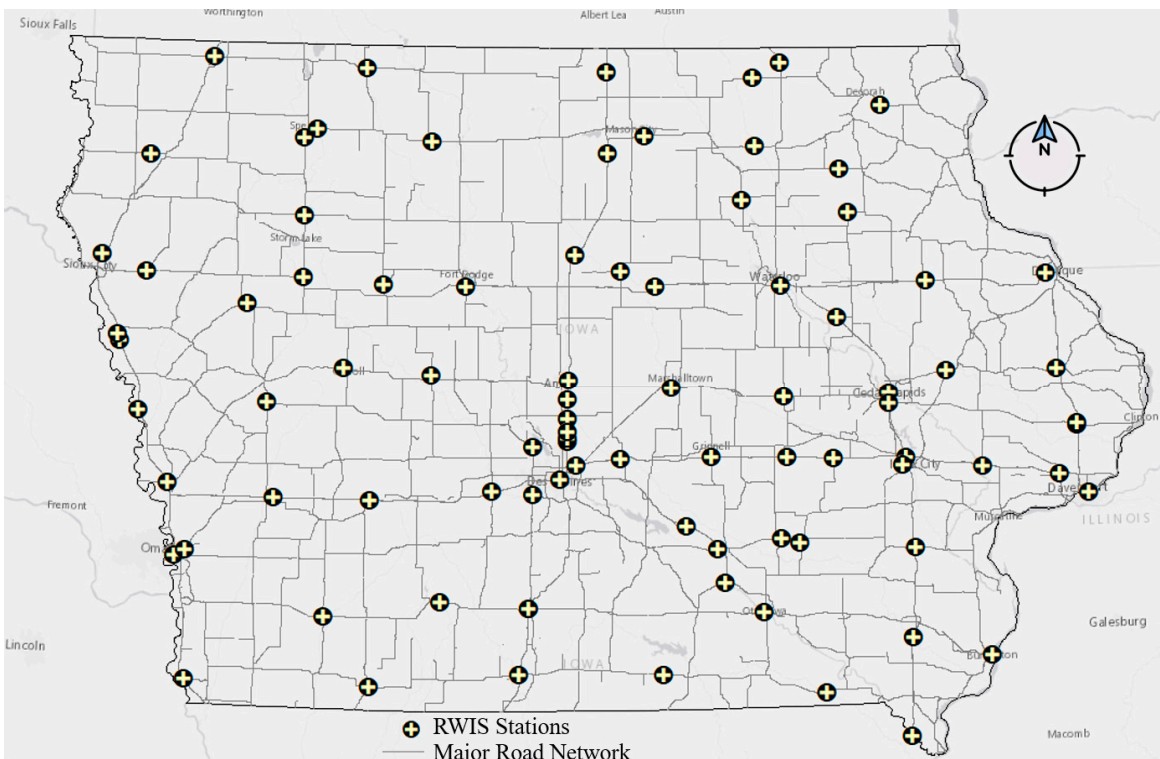

**Figure 2.** RWIS network and major roads in the state of Iowa.

### 3.2. Data Description and Integration

The RWIS data used in this study was downloaded from Iowa State University's website (http://mesonet.agron.iastate.edu/RWIS/ (accessed on 30 November 2017)). Variables that were recorded includes air and surface temperature, dew point temperature, visibility, wind speed, road surface conditions, etc., collected at 15 to 20-min intervals. Winter season data (October 2016 to March 2017) was processed based on the quality check procedures discussed in the Section 2. Among these various RWIS measurements, road surface temperature (RST) was considered to be the most critical as it has a significant influence on the formation of ice and road surface friction, both of which are crucial factors for winter road maintenance (WRM) operations [30]. Post-processing was done using the R statistical package—version 3.2.5 [31,32] for the semivariogram analysis. Here, spatial and temporal semivariograms were constructed by considering space and time attributes separately. The output variograms (spatial and temporal) were then combined into a joint semivariogram using spatiotemporal anisotropy parameters (StAni), allowing us to preserve both spatial and temporal features. StAni represents the number of space units equivalent to one time unit. In this study, joint semivariograms for a mid-winter month were utilized for kriging variance determination. The continuity ranges of autocorrelation are presented in Figure 3. The spatial range of the variable of interest (RST) was found to be around 20 km for the month of January, while the temporal range was as approximately 21.5 h. The resultant joint semivariogram range was found to be 17 km in this case, which is lower than the spatial range. This finding makes intuitive sense since both spatial and temporal attributes are preserved in the joint semivariogram. The readers are referred to our previous work for a detailed investigation on joint semivariogram analysis for multiple weather variables [19].

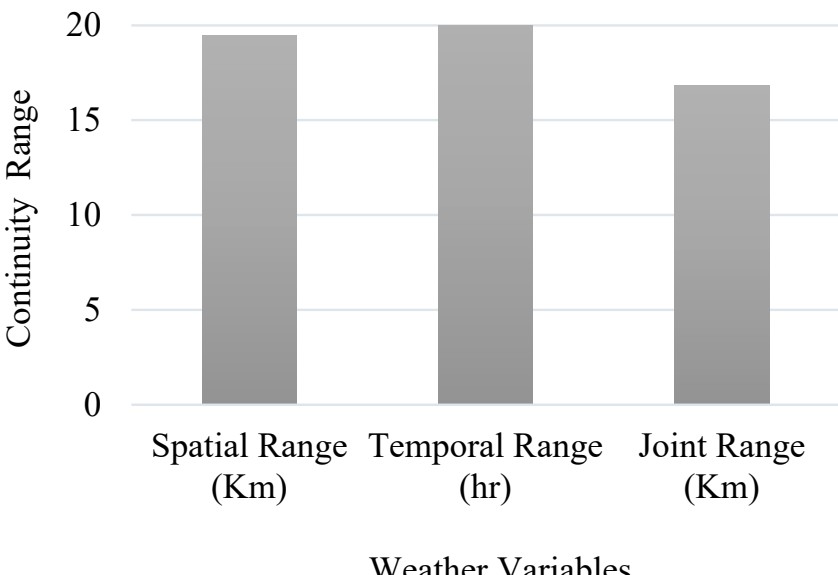

**Figure 3.** Spatial, temporal, and joint semivariogram parameters of RST for January 2017.

The parameters of the joint semivariogram were used in this study to evaluate the impact of RWIS configurations on *NCI*. The state of Iowa was used as the experimental boundary for determining the kriging variance. In addition, the major road network of this state was used as a constraint as to where the kriging estimation will be conducted; meaning that the observed RWIS measurements were used to estimate the unsampled location that lies on the major road network of Iowa. State boundary and major road network shapefiles were integrated within ArcGIS [33] to create a 5 km × 5 km grid surface of unsampled locations for what the kriging estimations were generated for. Afterwards, the variance was translated into *NCI* for the impact assessment of the RWIS network. The following section discusses the findings of the analysis.

## 4. Results and Discussion

This study focuses on quantifying the benefit of optimal RWIS network by evaluating the collision reduction potential of various RWIS configurations. In our previous study, the methodological framework for determining the optimal RWIS network was based on the concept that every location solution or RWIS configuration is associated with an objective function value (kriging variance). The optimal location solution has the lowest objective function value, and it is assumed to provide the maximum network monitoring coverage. In this study, the RWIS network coverage index (*NCI*) was determined for a set of RWIS configurations to establish a link (if it exits) between *NCI* and safety benefits. Kriging estimation error was used here to determine the *NCI*, while percent collision reduction was used as a performance indicator to quantify its benefit. The findings of this study are described below:

### 4.1. Dependency of NCI on RWIS Configuration

The relationship between kriging variance and *NCI* can be derived from the concept that *NCI* is inversely related to kriging variance. Hence, a proportional factor should be introduced to construct the relationship as defined previously in Equation (5).

It was assumed that an optimal RWIS density provides full network coverage of Iowa with an *NCI* value of 1. According to one of our previous studies, the optimal number of RWIS stations for Iowa is 61 [18]. Hence, at best, '*K*' in Equation (5) is equal to the kriging variance associated with this optimal number, and the maximum number of RWIS stations is capped at 61 because kriging variance cannot, or at least theoretically, go below optimal. Kriging variance is calculated using the joint semivariogram model developed in our previous study through a series of geostatistical analyses, where both spatial and temporal aspects were preserved [19]. Here the estimation variance was determined for an increasing number of RWIS stations. As the number of RWIS stations increases, the monitoring capability is expected to improve. This phenomenon is represented in Figure 4 by the decrease in kriging variance as the number of stations increases.

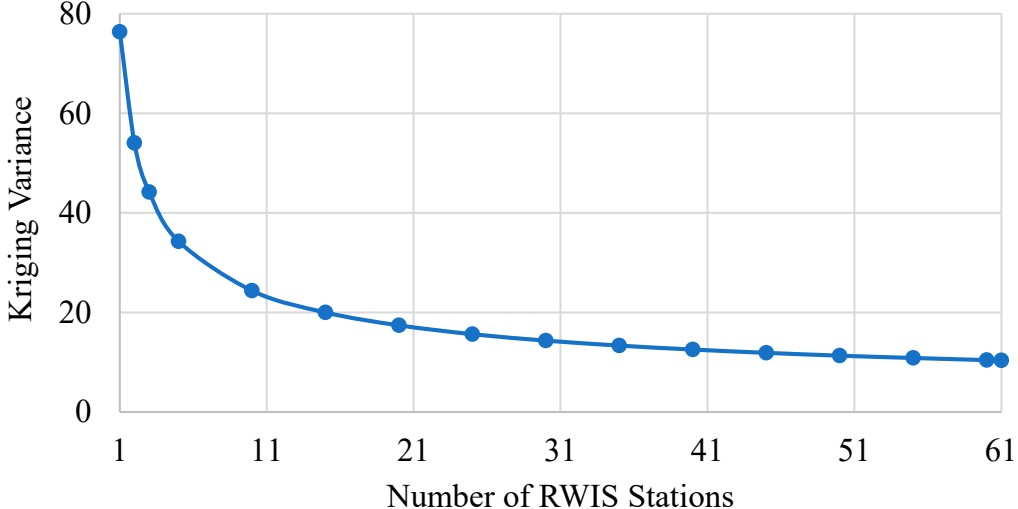

**Figure 4.** Plot of kriging variance for different number of RWIS stations.

From Figure 4, the value of kriging variance associated with the optimal scenario is 10.36—the number at which the greatest rate of change on kriging variance happens to occur. At this point, the full monitoring coverage can be achieved with an *NCI* value of 1. Thus, the proportional factor, *K* = 10.36 is used to update the equation as follow.

$$NCI = \frac{10.36}{Kriging\ Variance} \qquad (11)$$

*NCI* values for different RWIS configurations can be achieved using Equation (11), which changes Figure 4 to Figure 5. According to Figure 5, the monitoring coverage increases as the number of RWIS stations increases. In contrast, the marginal benefit gained with each additional RWIS decreases. The combination of these two effects results in the graph being concave shaped.

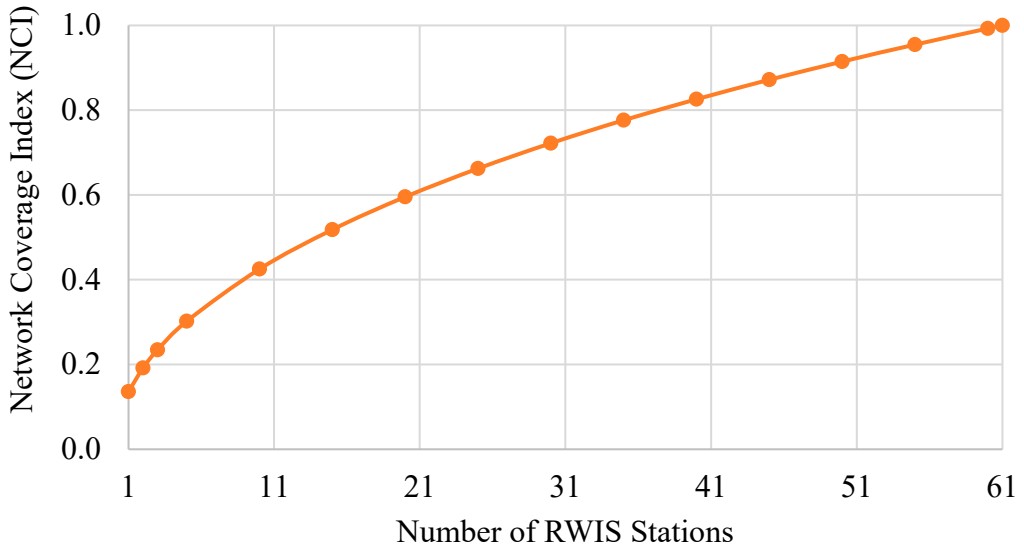

**Figure 5.** Plot of *NCI* for different number of RWIS stations.

*NCI* and kriging variance for different RWIS configurations is presented in Figure 6 to demonstrate how monitoring coverage changes with an increase in the number of stations. An example of this is the difference between scenarios one and six. Only 30% (*NCI* = 0.3) of optimal coverage could be provided as a result of having only five stations. By contrast, due to the increase in the number of stations in scenario six, the coverage level increased to 70%. Furthermore, the scenario with 5 RWIS stations generated an estimation error of 34.29, while a much smaller value (14.35) is obtained from the 30 stations scenario. It is clear from the above discussion that the *NCI* strongly depends on the density of the RWIS network. Thus, *NCI* is used in the subsequent section as a performance indicator to determine traffic safety benefits.

### 4.2. Impact Assessment of Iowa's RWIS Network

Our recent study examined the safety benefits of RWIS stations in Iowa using before-and-after Empirical Bayes (E.B.) method [22]. This method requires collision data before and after the implementation of the countermeasure. The study period was isolated to 2008–2019 and according to the operation information of the RWIS stations of Iowa, 30 stations were implemented within the study period. The selected 30 stations were filtered using a review criterion including data review, geometry review and operation review as discussed previously. This study considered 2 years of only winter months, (i.e., November to March) of before implementation and after implementation periods in the analysis. Thus, stations that did not meet this requirement for inadequate data sizes or shorter operational period, were removed from the analysis. Secondly, geometric changes near RWIS stations during the before and after period were reviewed to identify major construction activities within the study period. Stations near major geometric changes were removed from analysis as variations in road geometry can lead to unexpected changes in collision behaviour. Lastly, operational issues associated with stations were identified by assessing the frequency of data collection. Stations having issues with data collection was considered to have negligible effect on WRM because of lack of information provided and removed from the analysis. At the end of the review, 11 out of 30 stations were eliminated

from analysis. Of the remaining stations, 7 stations along with associated service area and treatment sites were selected and are presented in Figure 7.

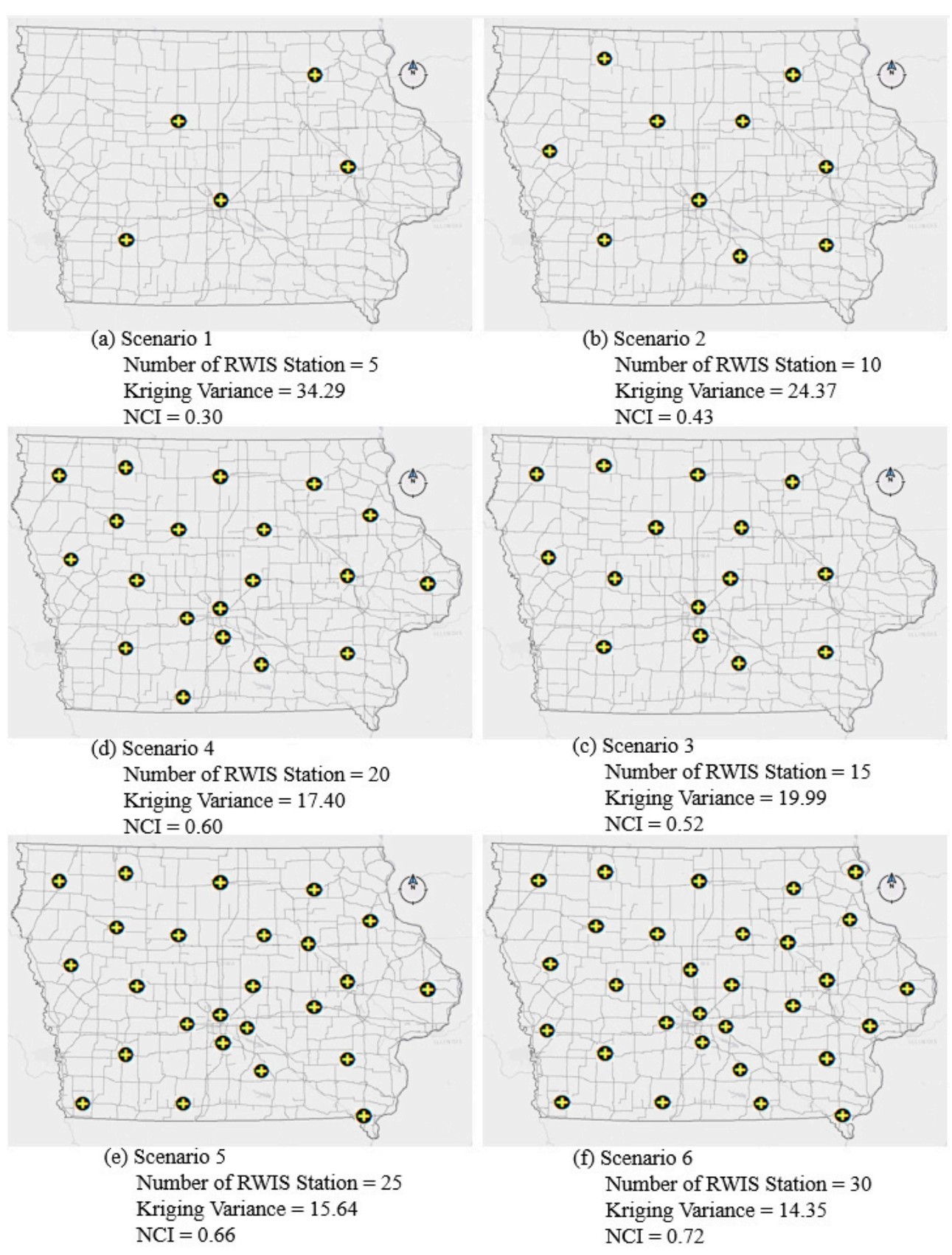

(a) Scenario 1
Number of RWIS Station = 5
Kriging Variance = 34.29
NCI = 0.30

(b) Scenario 2
Number of RWIS Station = 10
Kriging Variance = 24.37
NCI = 0.43

(d) Scenario 4
Number of RWIS Station = 20
Kriging Variance = 17.40
NCI = 0.60

(c) Scenario 3
Number of RWIS Station = 15
Kriging Variance = 19.99
NCI = 0.52

(e) Scenario 5
Number of RWIS Station = 25
Kriging Variance = 15.64
NCI = 0.66

(f) Scenario 6
Number of RWIS Station = 30
Kriging Variance = 14.35
NCI = 0.72

**Figure 6.** *NCI and kriging variance for different scenario.*

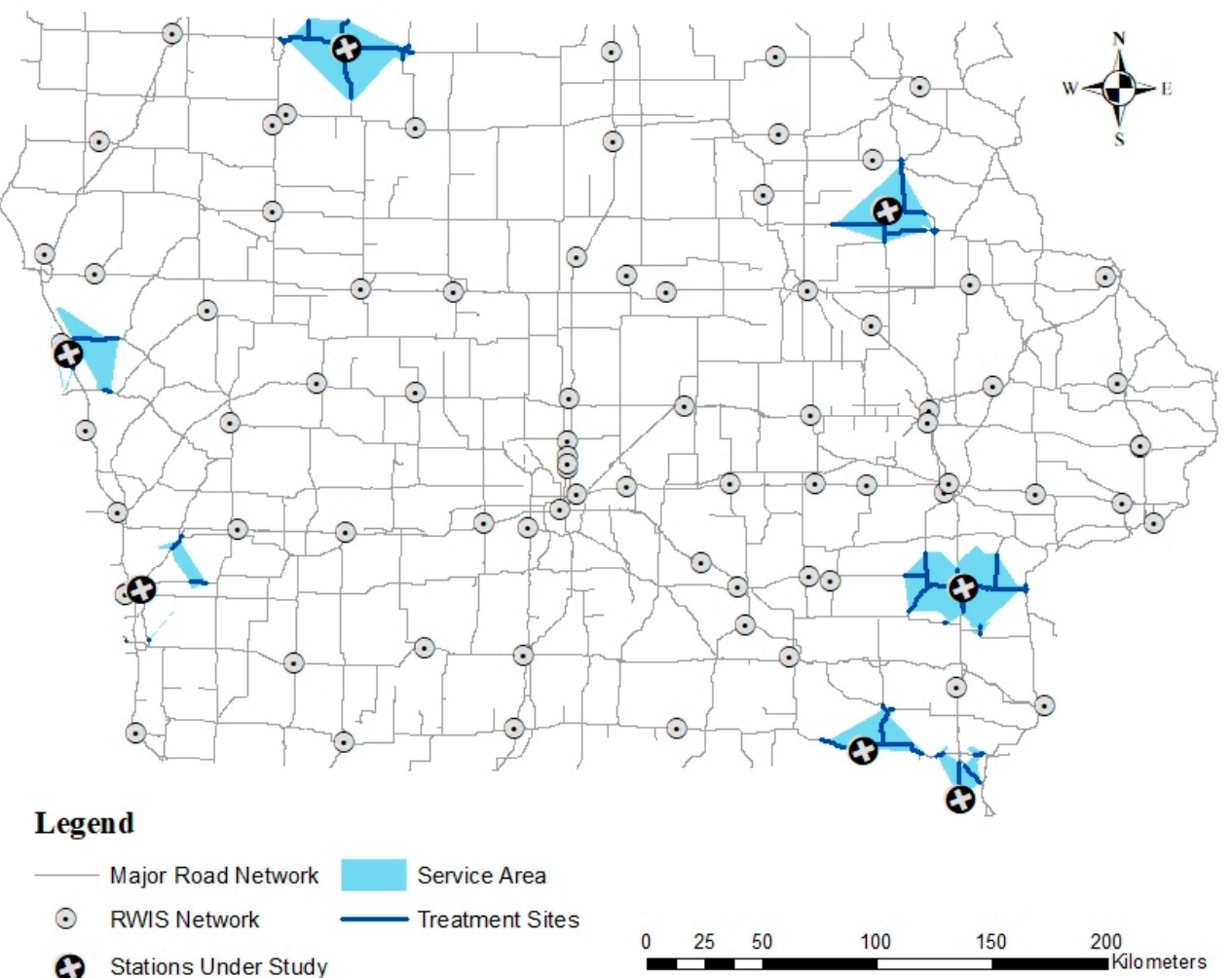

**Figure 7.** Map of Iowa showing various elements used in safety evaluation study.

Effect of a countermeasure (RWIS station) can be assessed by observing the change in collision frequency for a number of sites that are under the influence of that RWIS station. Here, a 30 km radius around an RWIS station was assumed as the influence region and accessible roads within this distance from the facility were considered its service area. For a number of cases where multiple stations were implemented close to each other, the service area under one station will overlap the service area of another station. Such stations were also removed from the analysis to avoid selecting sites that could be under the influence of another station. At this stage, 12 RWIS stations were eliminated from the analysis because the influence regions for these stations were partially or completely overlapped with another station. Hence, 7 stations were selected for the safety evaluation that has a significant influence region with a reasonable number of sites.

After selection of the treatment sites, a group of reference sites were required for the local calibration of SPF. Here, a set of desired reference sites were extracted along with the site-specific information including road type and traffic volume. The site-specific information was cross-checked with treatment sites and a group of reference sites was selected that can represent all 7 treatment sites. The dataset from the reference sites were then used for model calibration of SPF and YCF, which is a critical part of EB analysis. The calibrated SPF equation is presented below:

$$\mu = e^{(-10.4861)} L^{(0.8246)} \cdot V^{(0.2291)} \tag{12}$$

The collision frequency predicted by SPF need to be adjusted using YCF to obtain more accurate prediction. Using the calibrated SPF, YCFs for the study period (2008 to 2019) were determined. The values of YCF from 2008 to 2019 are: 1.865, 1.189, 1.490, 1.009, 0.795, 0.884, 1.089, 0.779, 0.913, 0.537, 0.669 and 0.976.

After the calibration of SPF and YCF, safety effectiveness of RWIS station was evaluated. According to the analysis result, the collision reduction potential for an RWIS station varies from 31.53% to 88.23%, with an average reduction of 65% in winter weather collisions [22]. The number of collision reductions varies from 4.73 to 27.61, with an average collision reduction value of 15. Since the total number of stations in each site ranges from 4 to 22, we can divide the number of collisions reduced by the number of stations to quantify the safety benefit of an individual station—an average value of 1.06. Table 1 depicts the collision reduction potential at different sites.

**Table 1.** Average Collision Reduction Calculation Based on Safety Evaluation of Selected RWIS Stations.

| Station ID | Collision Reduction (%) | Number of Collision Reduction | Total Sites | Collision Reduction Per Site |
|---|---|---|---|---|
| RCCI4 | 59.49 | 5.4 | 7 | 0.7714 |
| RCLI4 | 83.11 | 8.81 | 12 | 0.7342 |
| RETI4 | 31.53 | 21.22 | 22 | 0.9645 |
| RSOI4 | 83.8 | 4.73 | 4 | 1.1825 |
| RAGI4 | 88.23 | 14.03 | 12 | 1.1692 |
| RAII4 | 46.87 | 27.61 | 19 | 1.4532 |
| RMYI4 | 63.35 | 22.93 | 20 | 1.1465 |
| Average = | 65.20 | 14.96 | - | 1.0602 |

By utilizing the average traffic safety benefit associated with RWIS stations, the collision reduction potential for each of the RWIS configuration is determined and plotted against the associated kriging variance as presented in Figure 8. According to Figure 8, error variance, which is an indicator of monitoring capability has strong effect on traffic safety. The dependency of collision reduction potential on kriging variance can be expressed with a power function as it is associated with the highest R-square value while fitting with a trendline. The relationship is presented in Equation (13). The output of this finding presents strong evidence that optimal RWIS locations, which is associated with minimized kriging variance, is able to provide superior transportation system (traffic safety) benefit.

$$\text{Collision Reduction Potential} = \frac{7777.6}{Kriging\ Variance^{2.059}} \tag{13}$$

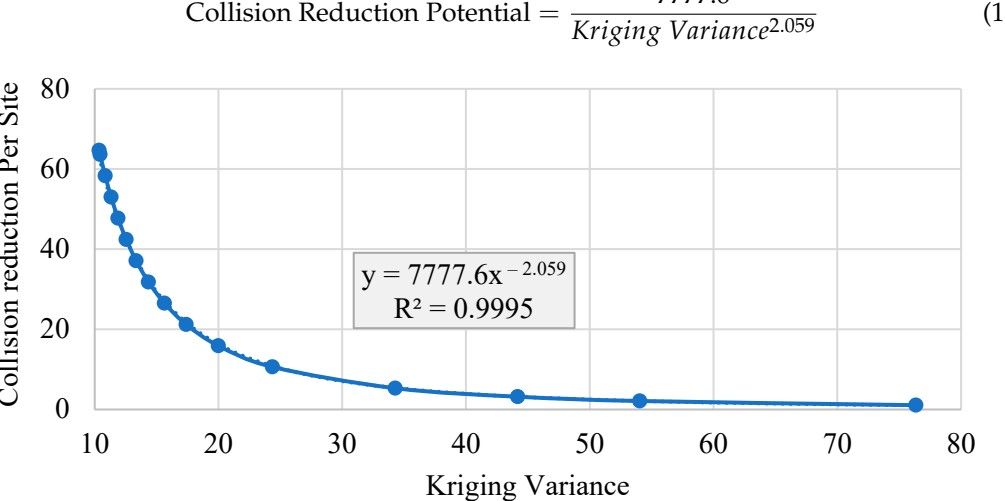

**Figure 8.** Plot of collision reduction potential with kriging variance.

In the last step, the dependency of safety effectiveness of RWIS network on *NCI* was determined by plotting it against the associated collision reduction potential (Figure 9). The findings revealed that *NCI* is highly correlated with collision reduction. An RWIS configuration with a higher *NCI* value was proven to be more effective for transportation safety than an RWIS network with a lower *NCI* value. For example, an RWIS network with 80% network monitoring coverage provides 40 collision reduction per site per year.

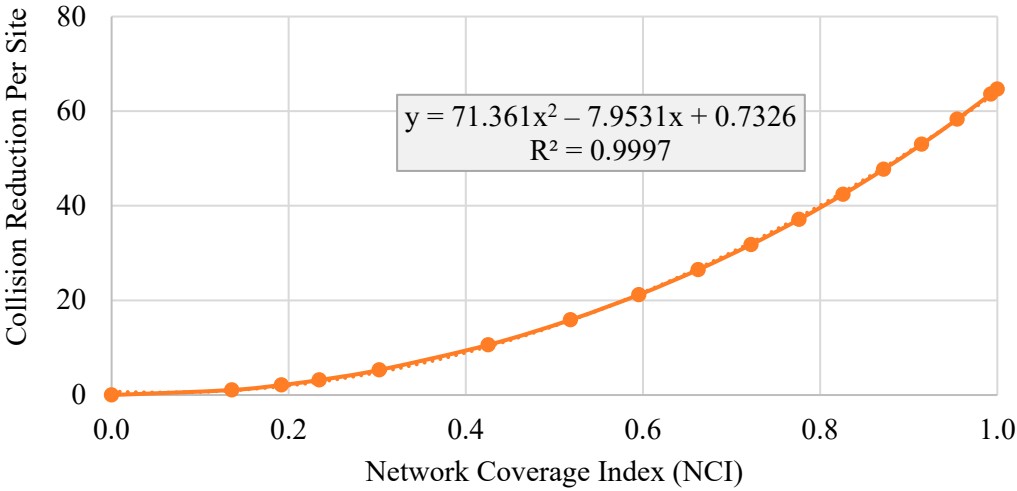

**Figure 9.** Plot of collision reduction potential with *NCI*.

The dependency of collision reduction potential on *NCI* can be expressed as a polynomial function in Equation (14) with a R-square value of 0.99. Here, polynomial function was used to fit trendline as it is associated with the highest R-square value.

$$\text{Collision Reduction Potential} = 71.36 \times NCI^2 - 7.95 \times \text{NCI} + 0.73 \tag{14}$$

It is evident from the above findings that an RWIS location solution with lower estimation error (higher *NCI*) will provide a significantly safer transportation network than another solution with higher estimation error (lower *NCI*). This result in turn justifies the previously developed location allocation strategy [19], where optimal RWIS location was selected based on lowest estimation error. It is clearly apparent that the optimal location solution is more beneficial in terms of safety effectiveness.

## 5. Conclusions and Recommendation

RWIS play a critical role in improving transportation safety, mobility, and winter road maintenance operations. Acknowledging their significant operational and environmental benefits, many North American transportation agencies have invested millions of dollars in deploying RWIS stations to strengthen the monitoring coverage of winter road surface conditions. To maximize the benefits of such systems, RWIS stations should be located systematically at a specific number of selected locations, which is referred as the optimal RWIS network. Our previous research provided a solid foundation for planning an optimal RWIS network. However, the goodness of the RWIS locations has never been examined, particularly the effect RWIS location solutions have on transportation safety. The key findings of this study are:

- The Network coverage index (*NCI*), a measure of monitoring capability, is intensely tied to the RWIS network configuration. A direct relationship between *NCI* and kriging variance has also been established in this study.

- Collision reduction potential of an RWIS network has been found to be proportional to and highly correlated with *NCI*. An RWIS configuration having higher *NCI* has a higher potential to reduce traffic collision, thus maximizing the safety effectiveness.
- The findings documented in this study concluded that the optimal RWIS locations, that are associated with lowest kriging variance (highest *NCI*) maximizes the overall benefit on transportation system.

  Recommendations for future study are given below:

- Study area of this research includes only the state of Iowa, which is a flatland area. Hence the benefit of optimal RWIS network should also be determined for other regions including hilly and mountainous regions for a complete and conclusive output.
- In addition to the traffic safety benefit, transportation mobility and winter road maintenance (WRM) benefits also need to be evaluated. One potential approach to determine mobility benefit could be based on AADT (Annual Average Daily Traffic) for a predefined coverage area before and after the installation of RWIS station. While the WRM benefit can be determined based on the maintenance cost for before and after-period of RWIS deployment.

**Author Contributions:** The authors confirm contribution to the paper as follows: study conception and design: S.B. and T.J.K.; data collection: S.B.; analysis, interpretation of results and draft manuscript preparation: S.B., D.S. and T.J.K. All authors have read and agreed to the published version of the manuscript.

**Funding:** This research was funded by Natural Sciences and Engineering Research Council of Canada (NSERC).

**Institutional Review Board Statement:** Not applicable.

**Informed Consent Statement:** Not applicable.

**Data Availability Statement:** The data presented in this study are available on request from the corresponding author.

**Acknowledgments:** The authors would like to thank the Iowa DOT for providing the data necessary to complete this research. This research is funded by the Natural Sciences and Engineering Research Council of Canada.

**Conflicts of Interest:** The authors declare no conflict of interest.

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
