# Peer review of "Safety Impact Assessment of Optimal RWIS Networks—An Empirical Examination"

_sustainability, doi:10.3390/su15010327_

Round 1
Reviewer 1 Report
Optimal RWIS network contributes to traffic safety. The authors develop network coverage index to evaluate RWIS network performance, then find the polynomial function relationship between NCI and safety effectiveness. This manuscript is not hard to follow, and I recommend some minor revision which need to be resolved.
1. There are a lot of citation errors in manuscript, like “Error! Reference source not found”, authors should check and fix them carefully
2. Equation (8) should be expressed as a generic form, and the parameter of Iowa case should be given at the Result section.
3. In figure 7, why the service area is far away from monitoring station which is located in the bottom and left?
4. How are the parameters determined in equation (12), Figure (8) and (9)?
5. The quality of figures in the manuscript need to be improved.
Reviewer 2 Report
This paper investigates the safety impacts of the Road Weather Information System (RWIS) using before-after analyses. It also studies the relationship between RWIS network coverage and safety; however, the results are somewhat obvious. If the implementation of RWIS has a positive safety effect, more stations (or higher network coverage) are accordingly associated with greater safety effects. Thus, the main contribution of this paper would be the safety component, which requires a lot more information. Much more literature on empirical Bayes (EB) and RWIS (or related systems) should be provided to support the claim in lines 117-118. The application of EB in this study must be presented in more detail, e.g., selection of treatment and non-treatment sites, estimations and model fit of the SPF functions and any other adjustment, e.g., YCFs. Figure 7 shows a strange pattern of treatment sites far away from the stations. Without a proper EB analysis, the results would be unreliable.
There are many typos and formatting errors in the paper.
Reviewer 3 Report
This manuscript reads well and conduct to identify safety impacts associated with the number of RWIS stations in a given region. For consideration for publication in this journal, recommend the authors modify this manuscript
- as you mentioned in this paper, the optimal RWIS stations was 61, which is the maximum number of the stations. So, this study assumes the RWIS stations cannot exceed the maximum number. To me, it seems like that more RWIS stations, less collision rate until a point of inflection. Then, the polynomial function may be changed.
- So, finally how many RWIS stations are needed in this region? I cannot find how to allocate these stations in the study region to provide adequate monitoring coverage? How did you make scenario?
- This study suggests 61 is the optimal solution of IOWA region after comparing to the safety impacts?
- minor : Error reference
